# Correcting the Correction: A Revised Formula to Estimate Partial Correlations between True Scores

Debra Wetcher-Hendricks



Department of Sociology and Anthropology, Moravian College, Bethlehem, PA 18018, USA; wetcher-hendricksd@moravian.edu; Tel.: +1-610-861-1415

**Abstract:** Bohrnstedt's (1969) attempt to derive a formula to compute the partial correlation coefficient and simultaneously correct for attenuation sought to simplify the process of performing each task separately. He suggested that his formula, developed from algebraic and psychometric manipulations of the partial correlation coefficient, produces a corrected partial correlation value. However, an algebraic error exists within his derivations. Consequently, the formula proposed by Bohrnstedt does not appropriately represent the value he intended it to estimate. By correcting the erroneous step and continuing the derivation based upon his proposed procedure, the steps outlined in this paper ultimately produce the formula that Bohrnstedt desired.

**Keywords:** classical test theory; classical true score theory; correction for attenuation; partial correlation coefficient

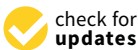



## 1. Introduction

Psychometricians have long acknowledged the presence and impact of measurement error. The very first principle of Classical Test Theory dictates that X = T + E. This equation suggests that an observed score (X) represents not only a measurement of the entity of interest, or the true score (T), but also some level of error (E) that distorts the researcher's observation. If researchers could determine the portion of an observed score that consists of the true score and of error, they could simply focus their analyses upon true-score values to obtain actual representations of the conditions they study. However, because error scores exist as latent components of observed scores, researchers cannot determine the exact extent to which they affect their data and the statistics computed from those data.

Despite the lack of such detailed information, some general understanding of the consequences that error scores can have upon statistics exists. Psychometricians know, for instance, that the presence of measurement error causes the correlation between observed scores to fall below that of true scores, a phenomenon known as attenuation [1]. However, the inability to obtain an actual true score or error score makes it impossible to determine the exact degree to which the observed score correlation underestimates the true score correlation.

This attenuation effect was first recognized by Spearman in 1904 [2]. Spearman, resigned to only estimate the correlation between true scores, proposed a formula to theoretically counteract the effects of attenuation on pairwise correlation coefficients. In his formula, $\rho_{T_X T_Y}$ represents an estimate of the Pearson correlation coefficient between true scores, the numerator contains the observed correlation coefficient, and the primed values in the denominator represent reliabilities. Applying the formula produces a value that exceeds the observed score correlation.

$$\rho_{T_X T_Y} = \frac{\rho_{XY}}{\rho_{XX'}\rho_{YY'}} \tag{1}$$

Spearman's correction for attenuation was groundbreaking with regard to Classical Test Theory. It provided increased accuracy in estimates of the relationship between

variables because it eliminated the effects of measurement error to the greatest extent possible. Consequently, esteemed statisticians and research analysts have given much credibility to Spearman's correction for attenuation [3].

One limitation of Spearman's formula, though, is its application only to pairwise situations. To address partial, part, or multiple correlations, one must apply the correction to each individual pairwise coefficient before inserting it into the relevant formula. This process becomes extremely tedious and time-consuming.

In response, Bohrnstedt [4] proposed a formula for a partial correlation coefficient containing an implicit correction for attenuation. (The partial correlation coefficient allows for consideration of the linear relationship between the independent and dependent variables, exclusive of any confounding effects that other factors may have upon these variables.) He had intended for his formula to allow the computation of an estimate for the partial correlation between true scores in a single step and his published explanations of his formula describe it as such. However, an algebraic mistake in Bohrnstedt's derivation makes the resulting formula incorrect.

## 2. Bohrnstedt's Derivation

Bohrnstedt bases his derivation on two psychometric equivalences. One states that dividing the covariance between two datasets by the product of the datasets' standard deviations produces the pairwise correlation coefficient, as shown in Equation (2).

This formula is one of many commonly used to calculate pairwise correlation coefficients.

$$\rho_{XY} = \frac{\sigma_{XY}}{\sigma_X \sigma_Y} \tag{2}$$

The other equivalence used by Borhnstedt allows him to adapt Equation (2) to partial correlations. It follows the leads of others [5,6], by focusing upon residuals between variables (e.g., $\sigma_{XZ}$). Simply, the covariances and standard deviations that appear in Equation (2) become covariances and standard deviations of residuals between these same variables and the potential confounding variable (Z). Bohrnstedt, therefore, defines the corrected partial correlation as the ratio between the covariance of estimated residuals and the square root of the product of the residuals' variances. These residuals, defined by relevant simple linear regressions, $X\text{-}b_{xz}Z$ and $Y\text{-}b_{YZ}Z$, become parts of Equation (3).

$$\rho_{T_X T_Y . T_Z} = \frac{\sigma_{(X - b_{XZ} Z)(Y - b_{YZ} Z)}}{\sqrt{\sigma^2_{X - b_{XZ} Z} \, \sigma^2_{Y - b_{YZ} Z}}} \tag{3}$$

The insertion of these expressions into the existing subscripts, as shown in Equation (3), indicates the consideration of multiple independent variables.

Expanding the covariance in Equation (4) and calculating the squared value of the terms in the denominator produces

$$\rho_{T_X T_Y . T_Z} = \frac{\sigma_{XY} - b_{YZ} \, \sigma_{XZ} - b_{XZ} \, \sigma_{YZ} + b_{YZ} \, b_{XZ} \, \sigma^2_Z}{\sqrt{(\sigma^2_X - 2b_{XZ} \, \sigma_{XZ} + \sigma^2_Z b^2_{XZ})(\sigma^2_Y - 2b_{YZ} \, \sigma_{YZ} + \sigma^2_Z b^2_{YZ})}} \tag{4}$$

Once again, following psychometric theory, Bohrnstedt redefines $b_{xz}$ as $\rho_{XZ}\sigma_X / \rho_{ZZ'}\sigma_Z$ and makes the comparable change for $b_{yz}$. This manipulation, along with restatement of each covariance as the product of the relevant correlation coefficient and standard deviations (e.g., $\sigma_{XY} = \sigma_X \sigma_Y \rho_{XY}$), produces

$$\rho_{T_X T_Y . T_Z} = \frac{\sigma_X \sigma_Y \rho_{XY} - \frac{\rho_{YZ}\sigma_Y}{\rho_{ZZ'}\sigma_Z}\sigma_X \sigma_Z \rho_{XZ} - \frac{\rho_{XZ}\sigma_X}{\rho_{ZZ'}\sigma_Z}\sigma_Y \sigma_Z \rho_{YZ} + (\frac{\rho_{YZ}\sigma_Y}{\rho_{ZZ'}\sigma_Z})(\frac{\rho_{XZ}\sigma_X}{\rho_{ZZ'}\sigma_Z})\sigma^2_Z}{\sqrt{(\sigma^2_X - \frac{2\rho^2_{XZ}\sigma^2_X}{\rho_{ZZ'}} + \frac{\rho^2_{XZ}\sigma^2_X}{\rho^2_{ZZ'}})(\sigma^2_Y - \frac{2\rho^2_{YZ}\sigma^2_Y}{\rho_{ZZ'}} + \frac{\rho^2_{YZ}\sigma^2_Y}{\rho^2_{ZZ'}})}} \tag{5}$$

A simplified version, results from adding and multiplying terms in Equation (5).

$$\rho_{T_X T_Y . T_Z} = \frac{\sigma_X \sigma_Y \rho_{XY} - \frac{2\rho_{XZ}\rho_{YZ}\sigma_X \sigma_Y}{\rho_{ZZ'}} + \frac{\rho_{XZ}\rho_{YZ}\sigma_X\sigma_Y}{\rho_{ZZ'}^2}}{\sqrt{\left(\sigma_X^2 - \frac{2\rho_{XZ}^2\sigma_X^2}{\rho_{ZZ'}} + \frac{\rho_{XZ}^2\sigma_X^2}{\rho_{ZZ'}^2}\right)\left(\sigma_Y^2 - \frac{2\rho_{YZ}^2\sigma_Y^2}{\rho_{ZZ'}} + \frac{\rho_{YZ}^2\sigma_Y^2}{\rho_{ZZ'}^2}\right)}} \tag{6}$$

Further algebraic simplifications involve three steps of factoring and combining terms. First, the standard deviations are factored within the numerator and the variances are factored within the denominator. Second, the existing terms within the numerator and within the denominator are added. The final step involves factoring reliabilities within the numerator and denominator as well as removing variances from beneath radical signs, making them standard deviations. These steps respectively appear as

$$\rho_{T_X T_Y . T_Z} = \frac{\sigma_X \sigma_Y \left(\rho_{XY} - \frac{2\rho_{XZ}\rho_{YZ}}{\rho_{ZZ'}} + \frac{\rho_{XZ}\rho_{YZ}}{\rho_{ZZ'}^2}\right)}{\sqrt{\sigma_X^2\left(1 - \frac{2\rho_{XZ}^2}{\rho_{ZZ'}} + \frac{\rho_{XZ}^2}{\rho_{ZZ'}^2}\right)\sigma_Y^2\left(1 - \frac{2\rho_{YZ}^2}{\rho_{ZZ'}} + \frac{\rho_{YZ}^2}{\rho_{ZZ'}^2}\right)}} \tag{7}$$

$$\rho_{T_X T_Y . T_Z} = \frac{\sigma_X \sigma_Y \left(\frac{\rho_{XY}\rho_{ZZ'}^2 - 2\rho_{XZ}\rho_{YZ}\rho_{ZZ'} + \rho_{XZ}\rho_{YZ}}{\rho_{ZZ'}^2}\right)}{\sqrt{\sigma_X^2\left(\frac{\rho_{ZZ'}^2 - 2\rho_{XZ}^2\rho_{ZZ'} + \rho_{XZ}^2}{\rho_{ZZ'}^2}\right)\sigma_Y^2\left(\frac{\rho_{ZZ'}^2 - 2\rho_{YZ}^2\rho_{ZZ'} + \rho_{YZ}^2}{\rho_{ZZ'}^2}\right)}} \tag{8}$$

and

$$\rho_{T_X T_Y . T_Z} = \frac{\frac{\rho_{ZZ'}(\rho_{XY}\rho_{ZZ'} - 2\rho_{XZ}\rho_{YZ}) + \rho_{XZ}\rho_{YZ}}{\rho_{ZZ'}^2}}{\sqrt{\left[\frac{\rho_{ZZ'}(\rho_{ZZ'} - 2\rho_{XZ}^2) + \rho_{XZ}^2}{\rho_{ZZ'}^2}\right]\left[\frac{\rho_{ZZ'}(\rho_{ZZ'} - 2\rho_{YZ}^2) + \rho_{YZ}^2}{\rho_{ZZ'}^2}\right]}} \tag{9}$$

It is this point in the derivation at which Bohrnstedt errs. His canceling of standard deviations within the numerator and denominator of Equation (9) is appropriate. However, he incorrectly cancels the reliability $\rho_{ZZ'}$ from the numerator and denominator, producing Equation (10).

$$\rho_{T_X T_Y . T_Z} \text{ “} = \text{” } \frac{\frac{(\rho_{XY}\rho_{ZZ'} - 2\rho_{XZ}\rho_{YZ}) + \rho_{XZ}\rho_{YZ}}{\rho_{ZZ'}}}{\sqrt{\left[\frac{\rho_{ZZ'} - 2\rho_{XZ}^2 + \rho_{XZ}^2}{\rho_{ZZ'}}\right]\left[\frac{\rho_{ZZ'} - 2\rho_{YZ}^2 + \rho_{YZ}^2}{\rho_{ZZ'}}\right]}} \tag{10}$$

Had $\rho_{ZZ'}$ appeared in all terms within the numerators of the general numerator and the general denominator, then he would have cancelled correctly. However, because only the first of the terms in these numerators contained $\rho_{ZZ'}$, such factoring and, consequently, such cancelling should not have occurred. Without realizing the flaws in Equation (10), Bohrnstedt continued with his derivation. He added components within the numerator and the denominator and then cancelled reliabilities from each. The equations

$$\rho_{T_X T_Y . T_Z} = \frac{\frac{\rho_{XY}\rho_{ZZ'} - \rho_{XZ}\rho_{YZ}}{\rho_{ZZ'}}}{\sqrt{\left[\frac{\rho_{ZZ'} - \rho_{XZ}^2}{\rho_{ZZ'}}\right]\left[\frac{\rho_{ZZ'} - \rho_{YZ}^2}{\rho_{ZZ'}}\right]}} \tag{11}$$

and

$$\rho_{T_X T_Y . T_Z} = \frac{\rho_{XY}\rho_{ZZ'} - \rho_{XZ}\rho_{YZ}}{\sqrt{(\rho_{ZZ'} - \rho_{XZ}^2)(\rho_{ZZ'} - \rho_{YZ}^2)}} \tag{12}$$

respectively, result.

The algebraic error leading to Equation (10) devalues Equation (12), proposed as the formula to compute a partial correlation coefficient corrected for attenuation. However, the equations presented before Equation (10) remain valid and can serve as a basis for fulfilling Bohrnstedt's original goal.

### 3. Correcting the Derivation

Using Bohrnstedt's reasoning and making the changes necessary to correct Equation (11) allows for the development of the formula that Bohrnstedt desired. Beginning the corrected derivation with Equation (9), squared reliabilities in the denominators of the general denominator can be removed from the radical sign. Cancelling this $\rho_{ZZ'}^2$ value with the same value in the denominator of the general numerator forms the single fraction

$$\rho_{T_X T_Y.T_Z} = \frac{\frac{\rho_{ZZ'}(\rho_{XY}\rho_{ZZ'} - 2\rho_{XZ}\rho_{YZ}) + \rho_{XZ}\rho_{YZ}}{\rho_{ZZ'}^2}}{\sqrt{\left[\frac{\rho_{ZZ'}(\rho_{ZZ'} - 2\rho_{XZ}^2) + \rho_{XZ}^2}{\rho_{ZZ'}^2}\right]\left[\frac{\rho_{ZZ'}(\rho_{ZZ'} - 2\rho_{YZ}^2) + \rho_{YZ}^2}{\rho_{ZZ'}^2}\right]}} \tag{13}$$

Distribution produces the final version of the formula:

$$\rho_{T_X T_Y.T_Z} = \frac{\rho_{XY}\rho_{ZZ'}^2 - 2\rho_{XZ}\rho_{YZ}\rho_{ZZ'} + \rho_{XZ}\rho_{YZ}}{\sqrt{\left[(\rho_{ZZ'}^2 - 2\rho_{XZ}^2\rho_{ZZ'} + \rho_{XZ}^2)\right]\left[(\rho_{ZZ'}^2 - 2\rho_{YZ}^2\rho_{ZZ'} + \rho_{YZ}^2)\right]}} \tag{14}$$

Equation (14) does exactly what Borhnstedt had suggested his formula would do. By inserting correlation and reliability coefficients into the appropriate positions, one can simultaneously compute a partial correlation coefficient and correct for attenuation.

### 4. Example

Evidence of Equation (14)'s effectiveness comes in the form of an example. Sabermetric data introduced by Wetcher-Hendricks (2006) in a different approach to calculating a partial correlation coefficient corrected for attenuation easily lend themselves to such an example. The data include values for walks ($X$), at bats ($Y$), and batting averages ($Z$) for key players on the New York Mets baseball team during the 2000 season. In particular, the observed correlation values, $\rho_{XY} = 0.811$, $\rho = 0.395$, and $\rho_{YZ} = 0.550$, and the reliability value of $\rho_{XX'} = 0.650$ from these data, fit into Equation (14). Then, a partial correlation coefficient corrected for attenuation of 0.7831 emerges through arithmetic simplification.

$$\rho_{T_X T_Y.T_Z} = \frac{(0.811)(0.650^2) - 2(0.395)(0.550)(0.650) + (0.395)(0.550)}{\sqrt{[(0.650^2) - 2(0.395^2)(0.650) + (0.395^2)][(0.650^2) - 2(0.550^2)(0.650) + (0.550^2)]}} \tag{15}$$

$$\rho_{T_X T_Y.T_Z} = \frac{0.2775}{\sqrt{(0.3757)(0.3317)}} = 0.7861 \tag{16}$$

This analysis used career means for walks, at bats, and batting averages as true scores, allowing for calculation of a true score partial correlation. A comparison between the corrected coefficient produced by Equation (14), the true-score partial correlation coefficient of 0.8983, and the observed score partial correlation coefficient of 0.7734 (Wetcher-Hendricks 2006) demonstrates the effectiveness of Equation (14). The observed score value differs from the true score value more than the value produced by Equation (14) does, indicating Equation (14)'s superiority in estimating $\rho_{T_X T_Y.T_Z}$.

Interestingly and importantly, in addition to lying closer to $\rho_{T_X T_Y.T_Z}$ than $\rho_{XY.Z}$ does, the value produced by Equation (14) improves upon the estimate obtained using the process described by Wetcher-Hendricks [7]. Thus, this equation provides the most effective means of estimating the partial correlation between true scores.

## 5. Conclusions

With Equation (14), Bohrnstedt has, albeit indirectly, achieved the goal of accounting for measurement error while estimating the partial correlation. He deserves credit for establishing the structure for the derivation of this equation; his minor algebraic oversight does not diminish the importance of his work.

This formula also has the same limitations as that offered by Bohrnstedt, and any formula that corrects for attenuation, does. Even Spearman's original correction for attenuation [2], which serves as a component in the derivation of Equation (14), relies upon reliability values. However, reliabilities generally remain unknown. Statisticians can do no better than to estimate these values. One must remember, therefore, that, like other equations that correct for attenuation, Equation (14) provides estimation, not exactness, with respect to the linear relationship it describes.

Nevertheless, the estimate produced by Equation (14) represents the true score partial correlation more accurately than the uncorrected correlation coefficient does. This improvement proves extremely valuable in applied research. For example, Gustafson [8] discusses its relevance in the field of epidemiology. He notes that those in this field often encounter mismeasured polychotomous variables, which includes continuous variables appropriate for regression analysis. Even small differences between a true score and an observed score can have dire consequences in the context of healthcare and medicine. Therefore, the ability to correct, as much as possible, for mismeasurement is extremely appealing. In fact, continued development of the formula in Equation (14) can further its usefulness in epidemiology as well as in other fields. Reasonable goals for future endeavors include deriving similar formulas for part and multiple correlations as well as adjusting these formulas to make them applicable to situations involving more than three variables.

Given the similarity between the partial and part correlation formulas, the procedure needed to derive a comparable formula for part correlations is likely very similar to that followed to obtain Equation (14). A formula to compute multiple correlation coefficients corrected for attenuation would not resemble the corrected partial coefficient quite as closely as the corrected part coefficient formula would. However, the existing derivation certainly suggests a general framework.

Expanding these formulas to manage more than three variables could follow patterns similar to those used in deriving the formulas for three-variable situations. However, as more variables become involved, the number of relationships between variables grows and, obviously, the complexity of the resulting formulas would increase with the addition of each variable.

Optimally, these developments could lead to formulas that correct for attenuation while computing partial, part, and multiple correlation coefficients for any number of variables. The generalizability of such formulas would make them highly useful for researchers wishing to analyze complex data.

**Funding:** This research received no external funding.

**Institutional Review Board Statement:** Ethical review and approval were not applicable for this study as it did not involve humans or animals.

**Data Availability Statement:** Data used for the example provided in this piece can be obtained by contacting the author.

**Conflicts of Interest:** The author declares no conflict of interest.

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
