# Peer review of "Correcting the Correction: A Revised Formula to Estimate Partial Correlations between True Scores"

_psych, doi:10.3390/psych3010003_

Round 1

Reviewer 1 Report

The manuscript entitled by “Correcting the Correction: A Revised Formula to Estimate Partial Correlations between True Scores” is a relatively comprehensive research method based on partial correlation coefficient and attenuation. The authors have attempted to study the psycho-metrics evaluations and verified potential corrections for results. The proposed design has been well introduced. For further improving the quality of research, there are some minor concerns:

  • Lines 112-114, please indicate if uncertainty needs to be identify/evaluated.
  • Missing discussions. The authors can also provide examples to further verify the methodology. It can be clinical database available on websites for downloading and utilization.
  • Please add a new table to summarize potential measurement in psychological research field, e.g. gender/sex differences.  

Author Response

For ease of reference, I have numbered Reviewer 1's comments (1-3) and provided my response to each, according to its number, below. Please note that lines identified  in these points may differ from those originally identified by the reviewers as a result of recent additions to and deletions from the text.

  1. Based upon the context of the passage that Reviewer 1 noted, I assume that the "uncertainty" he or she mentions relates to the use of estimates, rather than exact values. I have added text (lines 144-150) to address this issue.
  2. Lines 114-138 now consist of a section entitled, "Example." The example provides verification of the derived formula's effectiveness.
  3. The table requested by Reviewer 1 does not appear in the text. I am unable to discern exactly what "potential measurement" variables would appear in such a table, even given his or her example of "gender/sex differences." Lines 151-160 (added in response to a comment from Reviewer 2) MIGHT provide a bit of this information. However, if Reviewer 1 strongly believes that additional information is needed, and can clarify his or her comment, I can provide further revisions.

Author Response

For ease of reference, I have numbered Reviewer 2's comments (1-8) and provided my response to each, according to its number, below. Please note that lines identified  in these points may differ from those originally identified by the reviewers as a result of recent additions to and deletions from the text.

  1. An explanation of the prime notation now appears in lines 35-39. Additionally, this explanation now appears before the relevant equation so that readers understand the notation before encountering the equation.
  2. As suggested, Equation 3 was removed and adjustments were made, evident in lines  70-71, to provide a logical transition from Equation 2 to the new Equation 3 (formerly Equation 4). Numbers of following equations were changed to reflect the removal of Equation 3.
  3. Equations. 6 and 7 (formerly Equations 7 and 8) have been corrected.
  4. Equation 10 (formerly Equation 11) now contains the "=" symbol suggested by Reviewer 2.
  5. As suggested, the former Equation 14 was removed. Reference to Equation 9, which is identical to the removed equation, appears in lines 104-107. Numbers of following equations were changed to reflect the removal of the former Equation 14.
  6. Equation 14 (formerly equation 16) has been corrected.
  7. The requested explanation appears in lines 144-150.
  8. The requested explanation appears in lines 151-160.